# FedCAFE: Federated Cross-Modal Hashing with Adaptive Feature Enhancement

## ABSTRACT

Deep Cross-Modal Hashing (CMH) has become one of the most popular solutions for cross-modal retrieval. Existing methods need to first collect data and then be trained with these accumulated data. However, in real world, data may be generated and possessed by different owners. Considering the concerns about privacy, data may not be shared or transmitted, leading to the failure of sufficient training of CMH. To solve the problem, we propose a new framework called Federated Cross-modal Hashing with Adaptive Feature Enhancement (FedCAFE). FedCAFE is a federated method which could use distributed data to train existing CMH methods under the privacy protection. To overcome the data heterogeneity challenge of distributed data and improve the generalization ability of global model, FedCAFE is endowed with a novel adaptive feature enhancement module and a new weighted aggregation strategy. Besides, it could fully utilize the rich global information carried in the global model to constrain the model during the local training process. We have conducted extensive experiments on four widely-used datasets in CMH domain with both IID and non-IID settings. The reported results demonstrate that the proposed FedCAFE achieves better performance than several state-of-the-art baselines. As the topic that training deep CMH in federated scenario is in its infancy, we plan to release the code and data to boost the development of the field. However, considering restriction of anonymous submission and size limitation, we could only upload the source code of FedCAFE as supplementary materials for peer review at the present stage.

## CCS CONCEPTS

• **Information systems** → *Information extraction*; *Information retrieval*.

## KEYWORDS

Deep Hashing, Cross-Modal Retrieval, Federated Learning

## 1 INTRODUCTION

Existing deep cross-modal hashing methods [3, 15, 19, 43] have achieved efficient approximate nearest neighbor search by mapping high-dimensional samples to hash codes in low-dimensional Hamming space [20, 25, 28, 34]. Such methods have obvious advantages in large-scale retrieval, providing high retrieval speed and satisfactory retrieval accuracy [24, 26, 48]. These methods often require a large amount of data for training, while in real-world applications, data is typically scattered across different devices or institutions. Collecting and centralizing a large amount of distributed data is not only expensive but also impractical. Considering the legal restrictions and growing concerns about data privacy protection, such training schemes may be forbidden as transmitting data may face potential security risks and threats.

Federated learning, as a distributed machine learning framework, can ensure that a global model is trained collaboratively by uniting a series of clients without sharing local raw data [12, 18]. The emergence of federated learning solves the problem that local data cannot be shared under privacy and security requirements. Therefore, using federated learning to accomplish distributed training for deep cross-modal hashing methods is a feasible solution, which could not only protect the privacy of each participant but also improve the global model's generalization ability and robustness. These advantages motivate this paper to propose a new federated cross-modal hashing framework.

As the clients come from different devices and have their own private data, the data distributions among different clients are likely to be inconsistent (not independent identically distributed, i.e., non-IID), leading to data heterogeneity, which is one of the fundamental challenges in federated learning. Non-IID problem among client data often affects the stability, convergence, and effectiveness of the global model to varying degrees. To address the issue of data heterogeneity, quite a few works have attempted to propose their solutions. Most of them try to limit the updates of clients. For example, FedProx [18] constrains the models of clients to be closer to the global model by adding a proximal term in local updates, and MOON [17] restricts the direction of local updates to reduce client drift. However, most of the existing federated learning algorithms are all designed and validated based on the image classification task. In this paper, we put our efforts into investigating the heavily understudied cross-modal hashing problem under federated learning settings. Compared with single-modal classification tasks, this research problem is more challenging. First, cross-modal retrieval tasks need to deal with multiple modalities of data samples simultaneously, such as image and text modalities. Second, in the retrieval process, it uses one modality sample (e.g. text) to retrieve another modality sample (e.g. image) with semantic similarity, which requires exploring the semantic relationship between different modality samples. Third, data heterogeneity and distribution imbalance problems in federated learning also need to be considered and solved.

In literature, existing researches on federated cross-modal retrieval methods are very limited. FedCMR [50] is the first attempt and is based on one deep cross-modal retrieval model. Considering that hashing methods have high retrieval efficiency and accuracy, deep cross-modal hashing has become one of the most popular solutions for cross-modal retrieval. Thus, a federated cross-modal hashing framework PLFedCMH [23] is proposed which dynamically achieves personalized parameter customization for different layers of each client by introducing a hyperparameter network on the server, to improve the performance of local hashing models. PT-FUCH [16] proposes a strategy for federated unsupervised cross-modal hashing task, which uses global prototypes to promote the alignment of client feature spaces. Although they have promoted the developments of federated cross-modal retrieval, there still exist

some problems which are not well solved. First, how to address data heterogeneity is crucial for federated learning and is still an open problem in federated cross-modal hashing domain. Second, discriminative multi-modal features play a fundamental role while existing methods fail to fully adapt the feature extraction of current cross-modal methods with distributed data. In deep cross-modal hashing retrieval tasks, it is essential to learn discriminative features that can correctly distinguish the similarity and differences between different modalities of samples.

To overcome the limitations, we propose a new framework called Federated Cross-Modal Hashing with Adaptive Feature Enhancement, which is also named FedCAFE for brevity. As a federated framework, FedCAFE is model-agnostic which means it could be used for any existing deep cross-modal hashing methods. The main parts of FedCAFE involve an adaptive feature enhancement module and a new weighted aggregation strategy. The module could learn better multi-modal features with the help of global semantic information, thereby constraining the raw features. With the new strategy, local models could be better aggregated into the global model whose generalization ability could be boosted. The major contributions are as follows:

- We propose a novel federated framework FedCAFE for training existing deep cross-modal hashing methods with distributed data. The framework could avoid transmitting private data of clients and ensure good generalization ability of the global model.
- We propose an adaptive feature enhancement module, which fully embeds global semantic information into feature learning. This module could ensure discriminative multi-modal features and lead to better hash learning. A new weighted aggregation strategy is also proposed. In addition, FedCAFE further utilizes the knowledge carried by the global model to constrain the training of local models. All of them endow the global model with enhanced generalization ability.
- We have conducted extensive experiments on four widely-used datasets. The proposed FedCAFE achieves better retrieval accuracy than all chosen baselines.
- As the studied topic is vital and promising, we will release the data and code for good reproducibility. Currently, the source code of FedCAFE is uploaded as supplementary materials.

## 2 RALATED WORK

**Cross-Modal Hashing.** In recent years, the development of Deep Neural Networks (DNNs) [5, 13, 32, 39] has brought a new wave of research to cross-modal hashing. On the one hand, compared with traditional methods [6, 21, 40], deep hashing methods can integrate feature learning and hash learning into an end-to-end framework, letting them promote each other and improve the quality of hash learning[9, 27, 35, 42]. On the other hand, due to the high nonlinear modeling capability, DNNs can adequately exploit the data distribution, so as to obtain satisfactory retrieval accuracy [1, 11, 22, 49]. Although existing deep cross-modal hashing methods have achieved great success, they may fail to handle the new requirements which are proposed by nowaday real-world applications. At present, data may be generated and possessed by different owners. Meanwhile, more and more concerns are raised about privacy protection.

Thus, the learning scheme of traditional cross-modal hashing methods, which first collecting all data and then using accumulated data to train a centralized model, becomes impractical as transmitting data may be unsafe and even forbidden. How to train the model with distributedly stored data becomes a tricky challenge. Fortunately, federated learning, which could use data in different silos to train a unified model under privacy protection, is proposed and introduced into cross-modal retrieval domain. FedCMR [50] is the first attempt to explore a deep cross-modal retrieval framework for distributed data. It proposes a federated learning framework and takes existing deep cross-modal retrieval method DSCMR [47] as an example to demonstrate the effectiveness. FedCMR proves the feasibility of deep cross-modal retrieval methods in federated learning scenarios. Afterwards, PLFedCMH [23] is proposed to combine federated learning with cross-modal hashing. It uses a server-side hypernetwork to achieve personalized parameter customization for different clients. PT-FUCH [16] explores unsupervised cross-modal hashing in distributed scenarios.

**Federated Learning.** When data distributions among clients in federated training are different, the local training process of each client will let the local model converge to local optimal. Meanwhile, non-IID data distribution may mislead the optimization direction of the global model, which will further affect the generalization ability of global model and even make global model difficult to converge. In order to alleviate such problems, there have been extensive works to contribute their efforts in different directions. For example, FedProx [18] adds a proximal term in local training process to make the updated local models more similar to global model. Similarly, MOON [17] adds regularization items in local training process to prevent local models drift. FedProto [38] normalizes the training of local models by restricting local category prototypes generated by each client to be closer to global prototypes. However, there are some limitations in this kind of methods. It violates the fact that optimal point of local empirical objective is different from global optimum, and reduces the performance and convergence speed of global model [4]. Some methods try to tackle the problems from another way, i.e., improving the aggregation algorithms to get better global model. For instance, FedNova [41] is a normalized method replacing the widely-used simple weighted aggregation to alleviate objective inconsistency. Personalized Federated Learning (PFL) presents a new solution to data heterogeneity among clients [7, 14, 29, 45]. Specifically, these methods generate a personalized model for each client that is suitable for local data distribution. $CD^2$-pFed [33] uses a channel decoupling technique guided by cyclic distillation, which promotes the collaboration between local weights and global weights at channel level to realize personalized local models. Although these improved methods achieve good convergence speed and performance, most of them only explore the performance of algorithms on classification tasks. For more challenging scenarios, such as cross-modal retrieval, directly using above methods are not optimal and even infeasible. Thus, in this paper, we focus on investigating the cross-modal retrieval task for distributed data based on federated learning.

# 3 METHOD

To support existing deep cross-modal hashing to be trained with distributed data, we propose a new framework FedCAFE. Please note that FedCAFE is a federated learning framework, not a specific hashing method. It can be integrated with existing hashing methods. Considering that most existing deep hashing methods have their own feature extraction backbone networks, we directly use these networks and build our FedCAFE upon the extracted features. The illustration of FedCAFE for local client training is presented in Figure 1. Details of each component and the overall algorithm of FedCAFE are described below.

## 3.1 Notations and Problem Statement

Without loss of generality, we use image and text modalities as examples to illustrate our federated deep cross-modal hashing framework.

In the federated learning setting, we assume that there are $K$ clients, each of which has its own private data. The multi-modal data on the $k$-th client can be represented as $D_k = \{x_i, y_i, l_i\}_{i=1}^{n_k}$, where $n_k$ denotes the number of image-text pairs, $x_i$ represents the $i$-th image, $y_i$ is the corresponding text, the label of the $i$-th sample is represented by $l_i \in \{0, 1\}^{1 \times C}$, and $C$ is the number of categories.

Most existing deep cross-modal hashing methods employ two separate backbone networks, one for image modalities and another for text. These networks are designed to extract features from original images and texts, respectively, which are then used for hash learning. Our FedCAFE is a federated learning framework designed to facilitate the training of deep cross-modal hashing in the federated learning scenario, rather than a specific hashing method. Thus, FedCAFE requires a selected deep cross-modal hashing method and directly utilizes its default architectures to extract features. This approach allows FedCAFE to seamlessly integrate with existing hashing methods, leveraging their established mechanisms for feature extraction. The extracted raw features are represented as $O_i^* \in \mathbb{R}^{1 \times r}$, where $r$ denotes the length of hash codes and $*$ indicates different modalities (gray area of Figure 1). Then, to better mine the semantic information, a classifier is added after the raw features, and the predicted labels for different modalities are represented as $\hat{l}_i^* \in \mathbb{R}^{1 \times C}$ (yellow area of Figure 1). FedCAFE tries to mitigate the effects of non-IID data distribution and improve the generalization ability of the global model by utilizing the proposed adaptive feature enhancement module to generate enhanced features and impose constraints on the raw features (green area of Figure 1). As both backbone networks and classifiers have parameters to be learned, we use $W_x$ and $W_y$ to represent the entire network parameters of the image and text modalities, respectively.

The objective of our algorithm is to update and optimize the parameters of the global model for each modality by federated learning to achieve optimal performance for distributed deep cross-modal hashing methods on retrieval tasks. The overall objective of the algorithm can be formulated as follows:

$$
\begin{aligned}
\min_{W_x} F(W_x) &= aggregate\left(\{F_k(W_x)\}_{k=1}^{K}\right) \\
\min_{W_y} F(W_y) &= aggregate\left(\{F_k(W_y)\}_{k=1}^{K}\right),
\end{aligned}
\tag{1}
$$

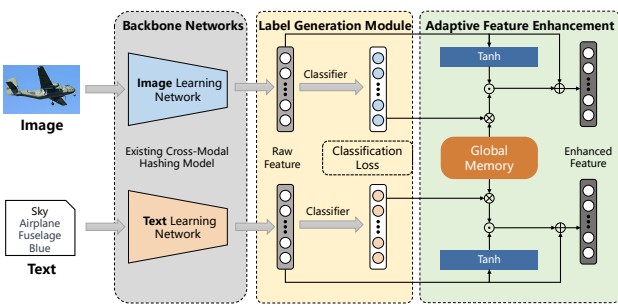

Figure 1: The illustration of FedCAFE for local client training. Gray area represents the raw feature extraction part, where the modality learning networks are the default settings of the chosen deep cross-modal hashing method. We further add a classification loss to make the learned features more semantic, which is illustrated in the yellow area. Green area denotes the proposed adaptive feature enhancement module, which outputs the learned enhanced features and utilizes them to constrain the raw features.

where $F_k(W_x)$ represents the loss function of image modality on the $k$-th client, and $F_k(W_y)$ is the loss function of text modality on the $k$-th client, $aggregate(\cdot)$ represents optimizing the global model parameters by minimizing the local loss of each client. More specifically, the loss functions for different modalities on the $k$-th client can be expressed as below:

$$
F_k(W_*) = \frac{1}{n_k} \sum_{i=1}^{n_k} \mathcal{L}_k(x_i, y_i, l_i; W_*).
\tag{2}
$$

As it is not feasible for the server to access local data of clients directly, we attempt to solve Eq. (1) in following manner. In the $t$-th communication round, the server sends the global model parameters of different modalities to $K$ clients. The corresponding local model parameters are overwritten with the global model parameters and used as the initial parameters of local model for the $t$-th round. The clients then train and update their local models using their own data, generating updated local model parameters for different modalities. Finally, the new local model parameters are uploaded to server for aggregation, generating global model parameters for next round.

## 3.2 Global Memory

Generally, in federated learning scenarios, the global model is trained on data that is distributed across multiple clients, and the update of global model needs to be transmitted to a central server for aggregation. One of the major challenges is that different clients may have significantly different data distributions, making it challenging to optimize a global model that performs well on all clients. To address this problem, a global memory rich in global semantic information is designed.

Specifically, in the $t$-th round of communication, the global memory is generated by aggregating the local memories of $K$ clients. For the $k$-th client, we first use the K-means algorithm to obtain local

memories of different modalities. These local memories serve as category prototypes of enhanced features. To reduce the domain gap among feature spaces of different modalities as much as possible, FedCAFE combines the local memories of the two modalities using the K-means algorithm to obtain the final local memory $P_k^t \in \mathbb{R}^{C \times r}$ for the $k$-th client in round $t$. This approach allows information from different modalities to fully integrate, promoting collaboration among modalities.

After obtaining the local memory for each client, the server further uses the K-means algorithm to aggregate them to generate the global memory $P_g^t$. The global memory aggregates the local memories of all participating clients and could better represent the feature distribution of the entire dataset. Therefore, we expect it can be used as a guide for individual client training, thereby alleviating the data heterogeneity problem among clients. It is worth noting that we do not violate the privacy policy of federated learning as we do not share raw data of clients.

The design of global memory could effectively promote information exchange and collaborative training among clients, thereby improving the generalization performance of local models on clients.

## 3.3 Adaptive Feature Enhancement Module

In federated learning, the data distributions among clients are generally non-IID, and clients can only learn knowledge from their local data, leading to a drift in the features learned by each client, which hinders the generalization of the global model. For retrieval tasks, the learned representations of the data are crucial. To learn better representations of multi-modal data, we hope the representations are as discriminative as possible. However, existing methods [23, 50] fail to consider how to enhance representations with the help of semantic information, which may make the learned representations not good enough.

Therefore, we propose a module that enables sharing of global semantic knowledge among clients. This means that even across different clients, as long as samples have the same semantics, the features generated will be more consistent. In addition to global memory, labels also contain rich semantic information, which can further promote knowledge sharing among clients. Specifically, by merging the predicted labels with the global memory, memory features rich in global semantic knowledge can be generated. Memory features can be expressed as:

$$V_M^* = \hat{L}^* \otimes P_g, \tag{3}$$

where $\hat{L}^*$ represents the predicted label matrix of local data and $*$ denotes different modalities, $P_g$ stands for the global memory. Using the matmul product $\otimes$ between them can produce memory features $V_M^*$.

Based on this, we introduce an adaptive selector $\mathbf{A} = Tanh\left(O^*\right)$ that integrates the original feature $O^*$ with the memory feature $V_M^*$ to enhance the original feature. The design of $\mathbf{A}$, i.e., using the $Tanh(\cdot)$, can not only directly obtain the weight of adaptive selector from original feature $O^*$, but also avoid the complicated adjustment of parameters. The enhanced feature $V_E^*$ is expressed as:

$$V_E^* = O^* + \mathbf{A} \odot V_M^*, \tag{4}$$

where $O^*$ represents the raw features extracted by backbone networks, $\odot$ is the Hadamard product. The enhanced features of image

modality can be represented as $V_E^x \in \mathbb{R}^{n_k \times r}$, and the enhanced features of text modality can be expressed as $V_E^y \in \mathbb{R}^{n_k \times r}$, typically, they are regarded as continuous surrogate of hash codes.

To more effectively constrain the raw features of samples, cosine similarity measurement is adopted to utilize enhanced features $V_E^*$ to constrain the original features $O^*$, thus improving the quality of feature representation. The formula is as follows:

$$\mathcal{L}_E = 1 - \cos\left(V_E^*, O^*\right), \tag{5}$$

where $\cos(\cdot)$ represents cosine similarity measurement.

In cross-modal retrieval tasks, the information contained in multi-modal data features is crucial as it can reveal the inherent correlations and complementary properties between different modalities. In order to fully utilize this information, FedCAFE further utilizes the global information carried in the global model to constrain the training of local models for each client. Specifically, when each client conducts local training, the features extracted by the global model are introduced as prior knowledge to provide guidance for the local training process. In this way, it ensures that local models can closely follow the guidance of the global model and better utilize the correlations between features of different modalities. The specific formula is as follows:

$$\mathcal{L}_C = \frac{1}{2}\left(1 - \cos\left(V_E^{*-k}, V_E^x\right)\right) + \frac{1}{2}\left(1 - \cos\left(V_E^{*-k}, V_E^y\right)\right), \tag{6}$$

where $V_E^{*-k}$ represents the enhanced features of different modalities on the $k$-th client, $*$ represents different modalities, $V_E^x$ represents the enhanced features of image modality when image sample passes through the corresponding global model, and $V_E^y$ represents the enhanced features of the text modality when the text sample passes through the global model of the corresponding modality. In this way, samples of different modalities can be effectively compared and matched in a unified feature space.

The information carried by the global model is obtained through training on multiple clients and can reflect the overall rules and patterns of data distribution. Transmitting this information to local models provides them with a broader and more comprehensive perspective, enabling them to generalize better to unknown data.

Since the learned hash codes not only need to maintain the similarity of the data in the feature space but also should have good classification ability, classification loss is added for supervised learning. FedCAFE utilizes KL divergence to measure the similarity between the predicted labels generated by classifiers and the true distribution of samples. For the training process of local models, the classification loss can be represented by the following formula:

$$\mathcal{L}_P = KL\left(\hat{L}^* \| L\right), \tag{7}$$

where $\hat{L}^*$ represents the predicted labels of different modalities, and $L$ represents the ground truth of samples.

## 3.4 Loss Function

The overall loss function can be constituted of hashing loss and federated loss:

$$\mathcal{L}_{all} = \mathcal{L}_{hash} + \alpha \mathcal{L}_E + \eta \mathcal{L}_C + \gamma \mathcal{L}_P, \tag{8}$$

where $\alpha$, $\eta$ and $\gamma$ are trade-off parameters. As the FedCAFE framework serves as a bridge to apply deep cross-modal hashing to the

federated learning scenario, hashing loss $\mathcal{L}_{hash}$ will not be specifically designed. As mentioned above, FedCAFE is model-agnostic which means it could be used for any existing deep cross-modal hashing methods. Thus, the objective loss of the chosen hashing method could be directly leveraged as $\mathcal{L}_{hash}$ in Eq. (8).

## 3.5 Global Model Aggregation with Memory

Data heterogeneity may lead to a decrease in model performance, as the local models on clients are trained on data that is not representative of the overall data distribution. To alleviate the problem of local model drift caused by data heterogeneity across clients and fully leverage the characteristics of deep cross-modal hashing retrieval tasks, we propose a novel weighted aggregation strategy to generate the global models of different modalities in each communication round on the server.

Specifically, we use the degree of difference between global memory and local memory as a measure of weight. To measure the similarity between the local memories of clients and the global memory, FedCAFE uses the negative log-likelihood function. The formula of the similarity between the local memory of $k$-th client and the global memory is:

$$Sim_k = -\sum_{i,j=1}^{C} \left( \mathbf{S}_{ij}\theta_{ij} - log\left(1 + e^{\theta_{ij}}\right)\right), \qquad (9)$$

where $\theta_{ij} = \frac{1}{2}(P_{k_{i.}}^t)(P_{g_{j.}}^t)^T$, and the local memory on the $k$-th client can be expressed as $P_k^t \in \mathbb{R}^{C \times r}$. We assume that the global memory and local memory of the same class should be similar, so we use the similarity matrix $\mathbf{S}$ to represent the relationship between them. If $P_{k_{i.}}^t$ and $P_{g_{j.}}^t$ are similar, $\mathbf{S}_{ij} = 1$, otherwise $\mathbf{S}_{ij} = 0$.

When the value of the negative log-likelihood function is larger, it indicates lower similarity between the local memory of $k$-th client and the global memory, suggesting that the specific information or knowledge contained on the $k$-th client is more unique compared to the global data distribution. Therefore, in order to fully emphasize these unique pieces of information and enhance the generalization ability of the global model, a higher weight is assigned to the model parameters on this client. The formula is as follows:

$$\alpha_k = \frac{e^{Sim_k}}{\sum_k^K e^{Sim_k}}, \qquad (10)$$

where $\alpha_k$ is the model weight of the $k$-th client. This weighting strategy helps ensure that the unique knowledge on the $k$-th client is adequately emphasized, while also aiding the global model in better learning and adapting to various data distributions.

Thus, the global model parameters for different modalities in round $(t + 1)$ can be expressed as:

$$W_x^{(t+1)} = \sum_{k=1}^{K} \alpha_k W_{x-k}^{(t)}$$
$$\qquad (11)$$
$$W_y^{(t+1)} = \sum_{k=1}^{K} \alpha_k W_{y-k}^{(t)},$$

where $W_{x-k}^{(t)}$ represents the local model parameters of the image modality on the $k$-th client in round $t$, $W_{y-k}^{(t)}$ is the local model

---

**Algorithm 1** FedCAFE

**Input:** communication rounds $T$, client numbers $K$, local epoch $E$, initial model parameters of different modalities: $W_x^0, W_y^0$

**Output:** $W_x^{(t+1)}, W_y^{(t+1)}$ and $P_g^{(t+1)}$

1: **Server executes**
2: **for** $t = 0, 1, \ldots, T - 1$ **do**
3:      **for** $k = 1, 2, \ldots, K$ **do**
4:          Send $W_x^t, W_y^t$, and global memory $P_g^t$ to client $k$
5:          $W_{x-k}^t, W_{y-k}^t, P_k^t \leftarrow$ LocalTraining $\left(k, W_x^t, W_y^t, P_g^t\right)$
6:      **end for**
7:      Compute fused global memory $P_g^{(t+1)}$
8:      Calculate global model parameters $W_*^{(t+1)}$ with Eq. (11)
9: **end for**
10: return $W_x^{(t+1)}, W_y^{(t+1)}$ and $P_g^{(t+1)}$
11: **LocalTraining**
12: **for** epoch $i = 1, 2, \ldots, E$ **do**
13:      **for** each batch of $D_k$ **do**
14:          Calculate enhanced features with Eq. (4)
15:          Train local model with overall loss Eq. (8)
16:      **end for**
17: **end for**
18: Compute local memory $P_k^t$
19: **return** $W_{*-k}^t$ and $P_k^t$ to server

---

parameters of the text modality on the $k$-th client in round $t$. The proposed weighted aggregation represents an innovative strategy that leverages the similarity between global and local memory to assess the difference between the data distribution of individual clients and the overall data distribution. By considering the contributions of each client, this mechanism effectively addresses the problem of local model drift resulting from non-IID data distribution and improves the generalization ability of the global model in federated learning.

## 3.6 Overall Algorithm

For better understanding of FedCAFE, we summarize the overall algorithm in Algorithm 1.

## 4 EXPERIMENTS

### 4.1 Datasets

We evaluated the proposed FedCAFE on four benchmark datasets for cross-modal hashing retrieval tasks, namely MIRFlickr-25K [8], NUS-WIDE [2], FashionVC [36], and Ssense [37]. The specific dataset partitioning strategy is shown in Table 1.

Table 1: The allocation of experimental datasets.

| Datasets | Volume | Training | Query | Retrieval |
|---|---|---|---|---|
| MIRFlickr-25K [8] | 20,015 | 10,000 | 2,000 | 18,015 |
| NUS-WIDE [2] | 195,834 | 10,500 | 2,100 | 193,734 |
| FashionVC [36] | 19,862 | 16,862 | 3,000 | 16,862 |
| Ssense [37] | 15,696 | 13,696 | 2,000 | 13,696 |

**Table 2: The MAP results of different federated learning methods on MIRFlickr-25K and NUS-WIDE.**

| Methods | MIRFlickr-25K | | | | | | NUS-WIDE | | | | | |
| | Image-to-Text | | | Text-to-Image | | | Image-to-Text | | | Text-to-Image | | |
| | 16bits | 32bits | 64bits | 16bits | 32bits | 64bits | 16bits | 32bits | 64bits | 16bits | 32bits | 64bits |
|---|---|---|---|---|---|---|---|---|---|---|---|---|
| Centralized [10] | 0.7383 | 0.7427 | 0.7527 | 0.7654 | 0.7669 | 0.7749 | 0.5903 | 0.6031 | 0.6093 | 0.6389 | 0.6511 | 0.6571 |
| Local [10] | 0.6405 | 0.6473 | 0.6544 | 0.6723 | 0.6821 | 0.6922 | 0.5042 | 0.5269 | 0.5418 | 0.4875 | 0.5025 | 0.5223 |
| FedAvg [30] | 0.6610 | 0.6734 | 0.6836 | 0.6907 | 0.7040 | 0.7143 | 0.5381 | 0.5581 | 0.5567 | 0.5186 | 0.5526 | 0.5609 |
| FedProx [18] | 0.6597 | 0.6724 | 0.6829 | 0.6899 | 0.7033 | 0.7133 | 0.5352 | 0.5565 | 0.5564 | 0.5176 | 0.5501 | 0.5599 |
| FedCMR [50] | 0.6729 | 0.6856 | 0.6956 | 0.7129 | 0.7229 | 0.7318 | 0.5572 | 0.5636 | 0.5735 | 0.5569 | 0.5781 | 0.5851 |
| MOON [17] | 0.6740 | 0.6862 | 0.6948 | 0.7081 | 0.7219 | 0.7317 | 0.5650 | 0.5840 | 0.5758 | 0.5664 | 0.5753 | 0.5867 |
| FedProto [38] | 0.6741 | 0.6892 | 0.6979 | 0.7138 | 0.7277 | 0.7392 | 0.5662 | 0.5863 | 0.6063 | 0.5695 | 0.5818 | 0.6051 |
| PLFedCMH [23] | 0.6531 | 0.6840 | 0.6984 | 0.6998 | 0.7221 | 0.7354 | 0.5738 | 0.5817 | 0.5852 | 0.5948 | 0.6028 | 0.6070 |
| FedCAFE | **0.7113** | **0.7206** | **0.7294** | **0.7451** | **0.7554** | **0.7648** | **0.5987** | **0.6425** | **0.6567** | **0.6065** | **0.6582** | **0.6617** |

**MIRFlickr-25K** contains 25,000 images collected from Flickr. It is multi-labeled, in which each sample is annotated with at least one of 24 semantic labels. Following the settings of [10], we removed the samples in which the number of tags is less than 20 and obtained 20,015 instances. In our experiments, we randomly selected 2,000 instances as the query set, left the remaining instances as the retrieval set, and randomly selected 10,000 samples from the retrieval set for training. This data partitioning strategy ensures that the model encounters diverse data distributions during both training and testing phases, thereby accurately evaluating its performance.

**NUS-WIDE** contains 269,648 images, each of which contains one or more of 81 labels. Following [10], we obtained 195,834 instances by using the 21 most frequent labels. In our experiments, we randomly picked up 2,100 instances as the query set, left the remaining samples as the retrieval set, and randomly chose 10,500 samples from the retrieval set for training.

**FashionVC** consists of 20,726 image-text pairs collected from the online fashion community Polyvore. Following [44], we removed labels with less than 25 samples and obtained 19,862 instances. Each sample has a hierarchical structure label, with 8 coarse-grained categories in the first level and 27 fine-grained categories in the second level. In our experiments, we leveraged the fine-grained labels and randomly selected 3,000 instances as query set, the remaining samples are used as retrieval set and training set.

**Ssense** consists of 25,947 image-text pairs collected from the online fashion community sense. Following the settings of [44], labels with less than 70 samples are removed and 15,696 instances are obtained. Similarly, the labels of this dataset are hierarchical, with 4 coarse-grained categories in the first layer and 28 fine-grained categories in the second layer. In our experiments, we selected fine-grained labels for annotating samples, and 2,000 instances are randomly selected as the query set, the remaining image-text pairs are used as the retrieval set and training set.

## 4.2 Implementation Details

**Backbone Architectures.** We adopted two existing deep cross-modal hashing retrieval frameworks as feature learning backbone networks for performance evaluation: 1) DCMH [10] for MIRFlickr-25K and NUS-WIDE; 2) SHDCH [44] for FashionVC and Ssense. We

employed the original codes of both of them as those codes have already been made publicly available by their authors.

**Baselines.** We compared our FedCAFE with several state-of-the-art baselines, including FedAvg [30], FedProx [18], MOON [17], FedProto [38], FedCMR [50], and PLFedCMH [23].

**Heterogeneity.** To investigate the effectiveness of FedCAFE with varying degrees of heterogeneity, we designed two settings: IID and non-IID. Since the MIRFlickr-25K and NUS-WIDE are multi-label datasets, it is difficult to simulate them as non-IID scenarios. Therefore, evaluations on these two datasets belong to the IID setting, where we randomly shuffled the training set and distributed it equally among the clients. For FashionVC and Ssense datasets, we followed previous works [31, 46] and used the Dirichlet distribution $\mathbf{Dir}(\beta)$ to simulate the non-IID scenarios, where a smaller $\beta$ indicates stronger heterogeneity among clients. In our experiments, we set $\beta$ to 0.5 and 0.2 for two situations.

**Hyperparameters.** For consistency, we set the communication rounds to 100 and the number of clients to 10 for all methods. For local training, the batch size is 128, the local training epoch is 20 on MIRFlickr-25K and NUS-WIDE, and 10 on FashionVC and Ssense. Additionally, we set the hyperparameter $\alpha$, $\eta$, and $\gamma$ of FedCAFE to 0.1, 0.1, and 1 respectively. For specific hyperparameters in baselines, we kept the same settings as their original papers.

**Evaluation Metrics.** We conducted two cross-modal retrieval tasks, which are retrieving texts using an image query (i.e., Image-to-Text) and retrieving images using a text query (i.e., Text-to-Image). We also conducted experiments on different lengths of hash codes. To evaluate the performance of all methods, we used Mean Average Precision (MAP), the most widely-used evaluation metric in cross-modal hashing retrieval methods. A higher MAP value indicates better performance. The baselines we compared have both traditional federated learning and personalized federated learning settings. For traditional federated learning, we evaluated the algorithm performance directly on the global model. For personalized federated learning, we evaluated the algorithm performance based on the average performance of the optimal local model on each client. To simulate practical personalized federated learning settings, we distributed the retrieval set and query set to clients according to their corresponding data heterogeneity and evaluated the learned models on the clients.

**Table 3: The MAP results of different federated learning methods with varying degrees of data heterogeneity on FashionVC.**

| Methods | $\beta = 0.5$ | | | | | | $\beta = 0.2$ | | | | | |
| | Image-to-Text | | | Text-to-Image | | | Image-to-Text | | | Text-to-Image | | |
| | 16bits | 32bits | 64bits | 16bits | 32bits | 64bits | 16bits | 32bits | 64bits | 16bits | 32bits | 64bits |
|---|---|---|---|---|---|---|---|---|---|---|---|---|
| Centralized [44] | 0.7620 | 0.7635 | 0.7616 | 0.9414 | 0.9527 | 0.9515 | 0.7620 | 0.7635 | 0.7616 | 0.9414 | 0.9527 | 0.9515 |
| Local [44] | 0.4006 | 0.4552 | 0.4585 | 0.3465 | 0.3928 | 0.3925 | 0.3128 | 0.3556 | 0.3639 | 0.2586 | 0.2837 | 0.2897 |
| FedAvg [30] | 0.5852 | 0.6753 | 0.7220 | 0.7368 | 0.8582 | 0.8976 | 0.4681 | 0.5502 | 0.6573 | 0.6321 | 0.7425 | 0.8204 |
| FedProx [18] | 0.5680 | 0.6615 | 0.7089 | 0.6837 | 0.8245 | 0.8623 | 0.4731 | 0.5777 | 0.6592 | 0.5171 | 0.6579 | 0.7580 |
| FedCMR [50] | 0.5868 | 0.6786 | 0.7228 | 0.7331 | 0.8652 | 0.8995 | 0.4745 | 0.5809 | 0.6587 | 0.6343 | 0.7514 | 0.8213 |
| MOON [17] | 0.5820 | 0.6808 | 0.7285 | 0.7024 | 0.7933 | 0.8638 | 0.4694 | 0.5722 | 0.6606 | 0.6399 | 0.6419 | 0.7311 |
| FedProto [38] | 0.6978 | 0.7099 | 0.7231 | 0.8808 | 0.9158 | 0.9030 | 0.5630 | 0.5785 | 0.5823 | 0.7663 | **0.8381** | 0.8109 |
| PLFedCMH [23] | 0.6027 | 0.6840 | 0.7198 | 0.7196 | 0.7726 | 0.8374 | 0.4974 | 0.5823 | 0.5808 | 0.5370 | 0.7583 | 0.7617 |
| FedCAFE | **0.7257** | **0.7603** | **0.7527** | **0.8935** | **0.9169** | **0.9093** | **0.6576** | **0.6840** | **0.7007** | **0.7903** | 0.8234 | **0.8310** |

**Table 4: The MAP results of different federated learning methods with varying degrees of data heterogeneity on Ssense.**

| Methods | $\beta = 0.5$ | | | | | | $\beta = 0.2$ | | | | | |
| | Image-to-Text | | | Text-to-Image | | | Image-to-Text | | | Text-to-Image | | |
| | 16bits | 32bits | 64bits | 16bits | 32bits | 64bits | 16bits | 32bits | 64bits | 16bits | 32bits | 64bits |
|---|---|---|---|---|---|---|---|---|---|---|---|---|
| Centralized [44] | 0.9692 | 0.9697 | 0.9720 | 0.9820 | 0.9881 | 0.9880 | 0.9692 | 0.9697 | 0.9720 | 0.9820 | 0.9881 | 0.9880 |
| Local [44] | 0.7161 | 0.7776 | 0.7895 | 0.6763 | 0.7436 | 0.7546 | 0.4888 | 0.5322 | 0.5518 | 0.4388 | 0.4752 | 0.4917 |
| FedAvg [30] | 0.8936 | 0.9517 | 0.9620 | 0.9386 | 0.9734 | 0.9840 | 0.7653 | 0.9059 | 0.9528 | 0.8028 | 0.9447 | 0.9792 |
| FedProx [18] | 0.8567 | 0.9424 | 0.9590 | 0.8789 | 0.9626 | 0.9773 | 0.7952 | 0.8964 | 0.9390 | 0.8152 | 0.9226 | 0.9715 |
| FedCMR [50] | 0.8953 | 0.9586 | 0.9631 | 0.9411 | 0.9763 | 0.9855 | 0.7956 | 0.9142 | 0.9537 | 0.8094 | 0.9459 | 0.9760 |
| MOON [17] | 0.9446 | 0.9628 | **0.9683** | 0.9549 | 0.9782 | 0.9845 | 0.7871 | 0.9205 | 0.9510 | 0.8028 | 0.9238 | 0.9806 |
| FedProto [38] | 0.9247 | 0.9620 | 0.9676 | 0.9390 | 0.9826 | 0.9845 | 0.9579 | 0.9585 | **0.9613** | 0.9789 | 0.9836 | 0.9825 |
| PLFedCMH [23] | 0.9545 | 0.9595 | 0.9557 | 0.9752 | 0.9827 | 0.9824 | 0.8832 | 0.8983 | 0.9014 | 0.8851 | 0.9366 | 0.9437 |
| FedCAFE | **0.9665** | **0.9671** | 0.9658 | **0.9856** | **0.9856** | **0.9860** | **0.9582** | **0.9595** | 0.9570 | **0.9801** | **0.9847** | **0.9832** |

## 4.3 Performance Comparisons

We conducted extensive experiments so as to evaluate the effectiveness of FedCAFE on retrieval accuracy.

**MAP Results in IID Scenarios**. Table 2 reports the retrieval performance of all compared algorithms on MIRFLICKR-25K and NUS-WIDE datasets. The first line reports the performance of centralized learning, the second line reports the performance of clients utilizing private data for local training. As shown in Table 2, Fed-CAFE achieves the best performance in all settings. Compared to the local training method without federated collaboration, federated learning methods have both shown performance improvements. For instance, on MIRFlickr-25K with the hash code length of 64 bits, FedCAFE outperformed the local training method by 7.5% in Image-to-Text task and 7.26% in Text-to-Image task, demonstrating the crucial role of federated collaboration during training. This aids models in acquiring stronger generalization capabilities, thereby enhancing overall performance. Additionally, compared to MOON, FedProto achieves higher retrieval accuracy, further proving the effectiveness of global prototypes, which enable clients to share global semantic information. However, in contrast, FedCAFE is more efficient and direct in enhancing sample features through the utilization of global memory.

To conclude, our proposed method FedCAFE is superior to all baselines in terms of accuracy.

**MAP Results in Non-IID Scenarios**. Table 3 reports the performance of all compared algorithms on FashionVC using different $\beta$ values. Table 4 lists the results of all methods on Ssense. In both tables, the first line reports the performance of centralized learning, the second line reports the performance of local training. From these tables, we can have following findings. (1) The proposed Fed-CAFE achieves the best or comparable performance in most cases on FashionVC and Ssense. (2) As data heterogeneity increases ($\beta$ becomes smaller), retrieval accuracy of all algorithms decreases, showing that non-IID distribution affects model learning and retrieval accuracy. (3) The MAP results of FedCAFE do not exhibit a significant decrease with the increase of Non-IID degree, demonstrating its stability and robustness in handling data distribution differences.

In a word, our method FedCAFE could also perform well in non-IID scenarios.

## 4.4 Ablation Analysis

**Validity of Functional Components.** To verify the necessity of each component proposed in FedCAFE, Table 5 shows the retrieval performance of FedCAFE after dropping some elements and losses

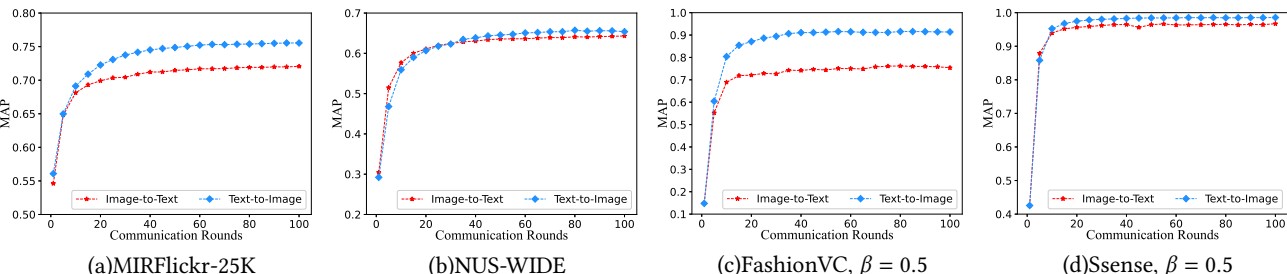

**Figure 2: Convergence curves of FedCAFE in 100 communication rounds on four datasets.**

**Table 5: Validation of the effectiveness of components in FedCAFE.**

| AFE | $\mathcal{L}_E$ | $\mathcal{L}_C$ | $\mathcal{L}_P$ | WA | Image-to-Text | Text-to-Image |
|-----|------|------|------|-----|---------------|---------------|
| – | – | – | – | – | 0.6753 | 0.8582 |
| √ | – | – | – | – | 0.6764 | 0.8592 |
| √ | √ | – | – | – | 0.6812 | 0.8624 |
| √ | √ | √ | – | – | 0.7488 | 0.9086 |
| √ | √ | √ | √ | – | 0.7548 | 0.9127 |
| √ | √ | √ | √ | √ | **0.7603** | **0.9169** |

in 32-bit hash code length on FashionVC with $\beta = 0.5$. In FedCAFE, there are two main elements: Adaptive Feature Enhancement module (AFE) and Weighted Aggregation strategy (WA), and three loss terms: feature constraint loss $\mathcal{L}_E$, cosine similarity loss $\mathcal{L}_C$, and classification loss $\mathcal{L}_P$.

As shown in Table 5, some observations can be found. (1) Fed-CAFE achieves the best performance when all components are employed. (2) Removing any component will reduce the performance. (3) FedCAFE will degrade into the conventional FedAvg when all components are removed. (4) By introducing the Adaptive Feature Enhancement module to enhance the raw features and using them to constrain the raw features, not only the drift of local models of each client be effectively alleviated, but also the performance of the global model can be significantly improved. (5) Deleting either $\mathcal{L}_E$ or $\mathcal{L}_C$ results in a decrease in retrieval performance, further highlighting the critical role of these two components. They can effectively unify the local features of clients into the global semantic space. To conclude, all components are necessary and effective.

**Hyperparameter Sensitivity Analysis.** To consider the impact of hyperparameter values, we further conducted experiments on the trade-off parameter $\alpha$, $\eta$, and $\gamma$ in 32-bit hash code length on FashionVC with the $\beta = 0.5$ data heterogeneity. We evaluated and analyzed the performance of FedCAFE by taking values for $\alpha, \eta, \gamma \in \{0.001, 0.01, 0.1, 1.0, 10, 100\}$. Figure 3(a) shows the impact of changing $\alpha$ on model performance when $\eta$ and $\gamma$ are fixed at 0.1 and 1.0, respectively. Similarly, Figure 3(b) shows the impact of changing $\eta$ on model performance when $\alpha$ and $\gamma$ are fixed at 0.1 and 1.0, respectively. Figure 3(c) shows the impact of changing $\gamma$ on model performance when $\alpha$ and $\eta$ are fixed at 0.1 and 0.1, respectively. It can be observed that when $\alpha, \eta, \gamma \in \{0.001, 0.01, 0.1, 1.0\}$,

even under different combinations of hyperparameters, the performance of FedCAFE remains relatively satisfactory. In our experiments, we set $\alpha$, $\eta$, and $\gamma$ as 0.1, 0.1, and 1.0, respectively.

**Communication Rounds Analysis.** In order to verify the impact of the number of communication rounds on FedCAFE, experiments were conducted separately on MIRFlickr-25K, NUS-WIDE, FashionVC, and Ssense with 32-bit hash code length. On FashionVC and Ssense, the Dirichlet parameter was set to $\beta = 0.5$. The experimental results are shown in Figure 2. It can be observed that as the number of communication rounds increases, the performance of the global model gradually improves and tends to stabilize. By increasing the number of communication rounds, each client can share and exchange knowledge thoroughly, thereby promoting the convergence of the model.

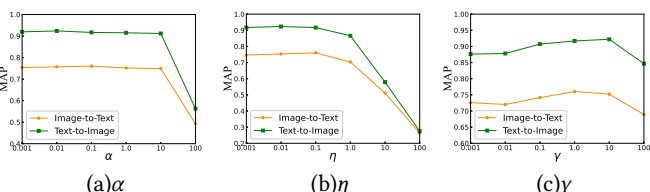

**Figure 3: The impact of different hyperparameter values on performance.**

## 5 CONCLUSION

In this paper, we propose a novel federated framework FedCAFE, which could train existing deep cross-modal hashing methods with distributed data while not sharing or transmitting private data on clients. FedCAFE uses default backbone networks to extract raw features of images and texts. Then, an adaptive feature enhancement module is proposed to utilize global semantic knowledge for enhancing and constraining the raw features. Furthermore, by leveraging the global knowledge carried in the global model, it can ensure that features with the same semantics across different clients are more unified. When aggregating local models on the server, a new weighted aggregation strategy is presented to improve the generalization capability of global model. Extensive experiments on four benchmarks demonstrate the effectiveness and convergence of the proposed FedCAFE. As the studied topic is vital and promising, we hope our work will promote further research in this field.

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
