# OpenReview forum: "FedCAFE: Federated Cross-Modal Hashing with Adaptive Feature Enhancement"
_acmmm.org/ACMMM/2024/Conference — MM2024 Poster_

### Official Review · Reviewer_H7jn · 2024-05-03

**Rating:** 3
**Confidence:** 3

**Summary:**

The paper proposes a Federated Cross-Modal Hashing method, FedCAFE, which includes an adaptive feature enhancement module and a new model aggregation strategy. Each client preserves local memory using the k-means method and uploads it to the server for further aggregation using k-means to obtain global memory. Ultimately, local memory is used to compute model similarity during server-side model aggregation, while global memory is employed to introduce global knowledge during client-side training. The authors validate the effectiveness of FedCAFE on four benchmark datasets.

**Strengths:**

1. This paper further validates the effectiveness of federated learning in the task of deep cross-modal hashing.
2. The article is written fluently, and the method's introduction is clear and easy to understand.

**Limitations:**

1. When deep cross-modal hashing is combined with federated learning, are there any new challenges that arise? The paper does not seem to address this. Intuitively, it seems that all federated learning methods could be applicable to this task.
2. Despite the authors emphasizing that FedCAFE can obtain more discriminative representations, they have not explained or validated why they are more discriminative.
3. The paper employs a parameter aggregation method based on similarity measurement, which is common in federated learning. The contribution of this paper lies in utilizing local memory for similarity measurement rather than proposing this aggregation strategy.
4. Does sharing memory between the server and clients raise privacy concerns? The paper needs further discussion on this issue.
5. In the ablation experiments in Table 5, when only the Adaptive Feature Enhancement module is added, the model performance hardly changes. The authors need more experiments to prove the effectiveness of AFE.

Other suggestions:
1. In the introduction, the description of contributions, the first and second points do not have a substantial difference, so there is no need to artificially separate them.
2. In the related work section, it is unnecessary to elaborate extensively on federated learning. Some space should be dedicated specifically to introducing "Federated Learning for Cross-Modal Hashing."

**Suitability:**

3

---

### Official Review · Reviewer_davG · 2024-05-10

**Rating:** 4
**Confidence:** 4

**Summary:**

This paper introduces FedCAFE, a federated framework designed to train deep cross-modal hashing methods using distributed data without sharing private client data. FedCAFE employs default backbone networks for feature extraction, followed by an adaptive feature enhancement module that utilizes global semantic knowledge to enhance and constrain raw features. It also utilizes a weighted aggregation strategy during model aggregation on the server to enhance the generalization capability of the global model.

**Strengths:**

1. The paper is well-structured and generally well-written.

2. The experimental results demonstrate significant performance improvements compared to the baseline methods.

3. The paper introduces an innovative concept, fully embedding global semantic information into feature learning, which is novel.

**Limitations:**

1.It's difficult to discern the relevance to federated learning in the flowchart depicted in Figure 1.

2 The manuscript still employs the methods of PLFedCMH and PT-FUCH, using feature prototypes to convey cross-modal semantics in the federated learning process, which seems lacking in innovation. The integration of federated learning with cross-modal retrieval raises the crucial issue of transferring cross-modal semantics without compromising user privacy, yet the paper appears to lack innovation in this regard.

3 The innovative aspect of the paper primarily lies in enhancing local models with global information. When global information is sufficiently learned, enhancing local models seems like a natural progression. However, the paper does not delve deeply into how to learn global information.

4 The paper mentions the transmission of local memories for two modalities, yet when local memories are updated, there is a risk of eavesdroppers capturing different local memories to obtain client-side private data, particularly concerning image data, which seems detrimental to federated learning.

**Suitability:**

2

---

### Official Review · Reviewer_kzB1 · 2024-05-20

**Rating:** 3
**Confidence:** 2

**Summary:**

This paper proposes a new framework called Federated Cross-modal Hashing with Adaptive Feature Enhancement (FedCAFE). FedCAFE is a federated method which could use distributed data to train existing CMH methods under the privacy protection. To overcome the data heterogeneity challenge of distributed data and improve the generalization ability of global model, FedCAFE is endowed with a novel adaptive feature enhancement module and a new weighted aggregation strategy. Besides, it could fully utilize the rich global information carried in the global model to constrain the model during the local training process.

**Strengths:**

1. This paper proposes a novel federated framework FedCAFE for training existing deep cross-modal hashing methods with dis-
tributed data.
2. The authors propose an adaptive feature enhancement module, which fully embeds global semantic information into feature learn-
ing. This module could ensure discriminative multi-modal features and lead to better hash learning. A new weighted aggregation strategy is also proposed.
3. The proposed FedCAFE achieves better retrieval accuracy than all chosen baselines.

**Limitations:**

I have some concerns regarding the experimental section.
1.	To enhance the credibility of the results, the author should consider involving more users instead of just 10.
2.	The authors should evaluate their method on more Non-i.i.d settings..
3.	How much computational overhead does the proposed method require? To demonstrate its practicality, a proper comparison of computational costs with existing methods should be provided.

**Suitability:**

3

---

### Official Review · Reviewer_diTb · 2024-05-24

**Rating:** 4
**Confidence:** 3

**Summary:**

This paper introduces a federated cross-modal hashing and adaptive feature enhancement framework called FedCAFE, which aims to solve the problem that existing cross-modal hashing methods are difficult to train in distributed data environments. Utilizing the federated learning framework, FedCAFE is able to collaboratively train global models across multiple clients without sharing raw data. By introducing an adaptive feature enhancement module and a new weighted aggregation strategy, the stability and convergence of the model when dealing with non-independently identically distributed data are improved, achieving better results than existing benchmarks on four widely used datasets. Search accuracy.

**Strengths:**

Innovative Federated Learning Framework: FedCAFE introduces a novel approach to federated learning by integrating cross-modal hashing with adaptive feature enhancement, which is a significant innovation in the field. Unlike traditional federated learning approaches that focus solely on a single data modality, FedCAFE effectively handles multiple modalities, ensuring that insights can be drawn from diverse data sources without the need to centralize the data. This ability to process and analyze data from different modalities federatively is particularly valuable for applications dealing with complex datasets like images and text where data privacy is a concern.

Robust Experimental Validation: The framework has been thoroughly tested across four widely-used datasets, demonstrating superior performance over existing benchmarks. This comprehensive validation not only shows FedCAFE’s effectiveness in enhancing retrieval accuracy but also proves its robustness in handling non-IID (independently and identically distributed) data across various client distributions. The experiments detailed in the paper include a range of scenarios, showcasing the framework's adaptability and scalability in real-world applications.

Practicality and Adaptability: FedCAFE’s design incorporates an adaptive feature enhancement mechanism that adjusts to the specific characteristics of the data it processes. This adaptability makes it highly practical for real-world applications where data characteristics can vary widely between different nodes in a federated network. Moreover, the inclusion of a new weighted aggregation strategy optimizes the learning process by prioritizing more informative updates from clients, which enhances the overall efficiency and effectiveness of the learning model.

Privacy-Preserving Capabilities: One of the significant advantages of FedCAFE is its ability to maintain high standards of data privacy. By design, the federated learning model ensures that sensitive data remains on local devices, and only essential information is shared across the network. This aspect is crucial for compliance with stringent data protection regulations and is highly relevant in sectors like healthcare and finance, where privacy concerns are paramount

**Limitations:**

Content of the article: First of all, I would like to know what the problem is with cross-modal hashing. Can you give an example? The second point is, in line 293, what is "+" and how to calculate it, whether to sum two vectors or to concatenate them. Hope you can answer me, thank you.

High Complexity of Implementation: One of the significant drawbacks of the FedCAFE framework is its complexity. The integration of cross-modal hashing with adaptive feature enhancements requires sophisticated algorithmic structuring and tuning, which may not only necessitate extensive computational resources but also a high level of expertise to implement effectively. This complexity could hinder the adoption of FedCAFE in environments where technical resources or expertise are limited, thus restricting its practicality in smaller organizations or institutions with constrained budgets.

Insufficient Generalization Validation: While the framework shows impressive results on the datasets used, the paper lacks a comprehensive validation across a broader array of scenarios that might encounter more varied data distributions or real-world conditions. The generalization capabilities of FedCAFE are crucial for its application in diverse environments, and without extensive testing in these contexts, there is a risk that the framework might not perform as expected when deployed under different or more challenging conditions than those presented in the study.

Dependency on High-Quality Input Data: The effectiveness of FedCAFE heavily relies on the quality and consistency of the input data. In federated environments, where data can originate from diverse sources with varying levels of quality and integrity, the performance of the framework could be significantly compromised. This reliance on high-quality input is a considerable limitation, as it might not always be feasible to ensure uniform data quality across all nodes in a federated learning setup.

Potential for Overfitting: The adaptive feature enhancement component, while beneficial for tailoring the learning process to specific data characteristics, also poses a risk of overfitting to the idiosyncrasies of the data available at individual nodes. This could lead to models that perform well on local or seen data but fail to generalize effectively to new or unseen data types or distributions. The paper could benefit from a more detailed discussion on strategies to mitigate this risk, ensuring the model remains robust and applicable to a wide range of data scenarios.

**Suitability:**

3

---

### Meta-Review · Area_Chair_PVEw · 2024-07-01

**Recommendation:** Accept (Poster)
**Confidence:** 4

**Metareview:**

This paper introduces a cross-modal hashing and adaptive feature enhancement framework called FedCAFE, which enable training  cross-modal hashing methods in distributed data environments. Utilizing the federated learning framework, FedCAFE is able to collaboratively train global models across multiple clients without sharing raw data. By introducing an adaptive feature enhancement module and a new weighted aggregation strategy, the stability and convergence of the model when dealing with non-independently identically distributed data are improved, achieving better results than existing benchmarks on four widely used datasets. Search accuracy.

The paper is well-written and conducted sufficient experiments to demonstrate the efficiency of the proposed method. The proposed method also has sufficient novelty on the integration of the feature and parameters across the client and servers in distributed environment.

The rebuttal materials has addressed the most problem proposed by reviewers. I checked the all the opinions and the answers from authors. The authors conducted experiments to prove the generalization ability of the framework that doubt by the reviewer.

Considering all the materials, I suggest present this paper which is well-written, clear, novel, comprehensive experimented in the Conference.